# Attribution Guided Distillation for Matryoshka Sparse Autoencoders

## Abstract

Sparse autoencoders (SAEs) aim to uncover monosemantic, interpretable features from deep networks but they face key limitations such as feature splitting and absorption. Matryoshka SAEs partially mitigate these issues with a nested dictionary architecture, yet they still suffer from redundancy, hyperparameter sensitivity, and higher reconstruction loss. To address these limitations, we propose a Distilled Matryoshka SAE (DMSAE) method that uses an attribution-guided selection strategy to iteratively identify and preserve high attribution neurons. In our method, these neurons are promoted into a frozen "core" that serves as the innermost prefix shared across all outer prefix reconstructions. When trained at layer 12 of Gemma-2-2B, DMSAE outperforms the MSAE baseline on most SAEBench benchmarks. DMSAE is model agnostic and may extend to other models where distilling a minimal core of features is desirable for interpretability and control.

## 1 Introduction

Large language models (LLMs) achieve state-of-the-art performance across many NLP tasks, yet their internal feature representations remain opaque. A useful working view is that high-level concepts correspond to approximately linear directions in a model's representation space so that behavior can be probed and steered with linear operations on those directions Wattenberg & Viégas (2024). Under this perspective, layer activations combine many underlying features, and understanding model behavior reduces to identifying and organizing those features. However, interactions among features and nonlinear gating make this far more complicated and serve as motivation for applying mechanistic interpretability to study LLMs.

Sparse autoencoders (SAEs) are a leading post-hoc tool for uncovering such features. SAEs learn an overcomplete, sparse latent dictionary that aims to map dense LLM activations into monosemantic units to improve interpretability and controllability Yan et al. (2024). In practice, sparsity alone is an imperfect proxy for interpretability, as users often observe feature splitting, feature absorption, and feature composition Bricken et al. (2023); Chanin et al. (2024); Leask et al. (2025). These pathologies can worsen as dictionaries scale, leading to redundancy and impaired benchmarks, despite improved reconstruction loss Karvonen et al. (2025); Gao et al. (2024).

To address hierarchy, *Matryoshka SAEs* (MSAEs) train nested dictionaries of increasing size, ensuring that early prefixes reconstruct general features and later prefixes add more details Bussmann et al. (2025); Zaigrajew et al. (2025). This helps preserve general features and delays specialization to larger prefixes. However, MSAEs still exhibit redundancy in deeper shells, sensitivity to hyperparameters, and higher reconstruction loss compared to other types of SAEs. Moreover, attribution across nested prefixes can be ambiguous, and simultaneous optimization of many prefixes can be computationally expensive.

We propose **Distilled Matryoshka SAEs (DMSAEs)**, an attribution-guided extension to MSAEs for LLM activations. DMSAE introduces a frozen core of neurons that is shared across all prefixes. Using attributions to the LLM's next-token loss, we iteratively promote high-attribution neurons from the first non-core prefix into this core. Encoder weights for promoted neurons are frozen to preserve their semantics, while decoder weights remain trainable to refine reconstructions.

In a proof-of-concept at layer 12 of Gemma-2-2B, DMSAE outperforms an MSAE baseline on most SAEBench metrics, improving AutoInterp, Absorption, SCR, and Sparse Probing while remaining

competitive on Loss Recovered and RAVEL. Because the method is model-agnostic and post-hoc, we expect it to also transfer to models from other scientific domains, where a dense and interpretable core is desirable for analysis and control.

## 2 BACKGROUND

### 2.1 SPARSE AUTOENCODERS AND MECHANISTIC INTERPRETABILITY

Mechanistic interpretability seeks to elucidate internal computations of deep neural networks by decomposing complex activations into simpler, semantically coherent units. A critical obstacle in this pursuit is neuron polysemanticity, where individual neurons respond to multiple unrelated concepts, hindering direct interpretation Olah et al. (2020); Elhage et al. (2022). Sparse Autoencoders (SAEs) address this by encoding dense activations into sparse, interpretable features via an encoder-decoder architecture:

$$\mathbf{f}(\mathbf{x}) = \sigma(\mathbf{W}^{\mathrm{enc}}\mathbf{x} + \mathbf{b}^{\mathrm{enc}}), \quad \hat{\mathbf{x}} = \mathbf{W}^{\mathrm{dec}}\mathbf{f}(\mathbf{x}) + \mathbf{b}^{\mathrm{dec}}, \tag{1}$$

where $\sigma$ enforces sparsity and non-negativity. Balancing reconstruction accuracy with sparsity encourages each neuron to represent distinct, monosemantic concepts Gao et al. (2024); Rajamanoharan et al. (2024). Nevertheless, standard SAEs face interpretability issues at scale, notably feature splitting, feature absorption, and feature composition, which obscure semantic clarity and reduce interpretability Bricken et al. (2023); Chanin et al. (2024); Leask et al. (2025).

### 2.2 HIERARCHICAL INTERPRETABILITY APPROACHES

Several approaches have been proposed to enhance hierarchical interpretability in sparse autoencoders by introducing structured and modular representations. Switch Sparse Autoencoders Mudide et al. (2025) introduce a Mixture-of-Experts (MoE) structure, partitioning feature spaces into specialized subnetworks. This conditional activation efficiently scales to large dictionaries by activating only subsets of features per input, improving computational efficiency. However, this method can also produce overlapping or ambiguous feature responsibilities across experts.

Universal Sparse Autoencoders (USAEs) Thasarathan et al. (2025) attempt cross-model feature alignment by learning a shared latent dictionary across multiple models. While this encourages reusable and generalizable features, it does not explicitly resolve inherent issues like redundancy or interpretability degradation with dictionary size growth.

Mixture-of-Experts for Intrinsic Interpretability (MoE-X) Yang et al. (2025) integrates interpretability directly into the model's architecture, promoting sparsity and internal modularity. Although this intrinsic interpretability approach significantly reduces polysemantic activations, it requires model architecture modifications and lacks flexibility for post-hoc analysis.

### 2.3 MATRYOSHKA SPARSE AUTOENCODERS AND HIERARCHICAL FEATURE LEARNING

Matryoshka Sparse Autoencoders (MSAEs) represent another significant hierarchical approach, extending conventional SAEs by employing a hierarchical nested training scheme, training multiple nested autoencoders simultaneously. In this method, multiple dictionaries of progressively larger sizes are optimized concurrently $m_1 < m_2 < \cdots < m_p$, each responsible for reconstructing inputs using only its subset of latent features Bussmann et al. (2025):

$$\hat{\mathbf{x}}_i = \mathbf{W}^{\mathrm{dec}}_{0:m_i}\mathbf{f}(\mathbf{x})_{0:m_i} + \mathbf{b}^{\mathrm{dec}}. \tag{2}$$

Early latent groups capture broad, general concepts because they are forced to reconstruct the data independently, without help from the more specialized features introduced later. Progressively larger latent groups then reconstruct inputs with increasing detail, adding specific features atop the general ones, thus preserving high-level abstractions while introducing fine-grained detail. This multi-level training setup ensures hierarchical interpretability by preventing later features from absorbing or fragmenting core concepts learned earlier Bussmann et al. (2025).

However, Matryoshka SAEs present several critical limitations. Empirical studies show that deeper latent shells often exhibit diminishing novelty, frequently encoding slight variations or fragments of earlier concepts rather than genuinely new, independent features. This redundancy results in many rarely-used neurons with subtle refinements, reducing overall interpretability. Additionally, the nested training scheme introduces ambiguity in feature attribution. Concepts that appear cleanly captured by a single latent in a smaller dictionary can fragment across multiple latents in larger dictionaries, complicating clear attribution and undermining the goal of strict monosemanticity. Moreover, the simultaneous training of multiple nested dictionaries substantially increases computational complexity and memory usage, significantly limiting scalability for practical applications with very large dictionaries Parsan et al. (2025); Leask et al. (2025).

### 2.4 ATTRIBUTION-GUIDED DISTILLATION FOR IMPROVED INTERPRETABILITY

To address these limitations, we propose an *attribution-guided core growth* approach that can incrementally build a dense and interpretable set of core neurons. Unlike methods relying on fixed or simultaneously trained nested dictionaries, our approach starts with a small initial core derived by promoting neurons from a previously trained Matryoshka SAE based on their reconstruction attribution scores. This core then serves as the innermost prefix in a newly trained Matryoshka SAE, in which core neurons remain permanently active (never sparsified) and contribute to reconstructions of all larger prefixes. In principle, this iterative process can continue iteratively to progressively distill a minimal core that can fully reconstruct the original model's representations.

We select the highest attribution neurons from the smallest prefix to attempt to preserve hierarchical structure, since larger prefixes encode more specific features. When promoted to the core, their encoder weights are frozen, while decoder weights remain trainable. As a result, each neuron continues to respond to the same features it encoded before promotion, while the decoder can refine how those features contribute to reconstruction. The goal of this approach is to reduce the constraints imposed by sparsity in Matryoshka SAEs and, with enough iterations and compute, a dense core could be constructed that is capable of an efficient and high quality reconstruction. We detail this methodology and a proof-of-concept study in the following sections.

## 3 METHODS

### 3.1 DISTILLED MATRYOSHKA SPARSE AUTOENCODER

Our proposed method, Distilled Matryoshka Sparse Autoencoder (DMSAE), extends the Matryoshka SAE framework introduced in Bussmann et al. (2025) and Zaigrajew et al. (2025), incorporating two key innovations: an explicit latent core of fixed size $c$ and an attribution-guided growth procedure (see Figure 1).

Let $\mathbf{x} \in \mathbb{R}^n$ be an input and define a sequence of nested latent sizes $\mathcal{M} = \{m_1, m_2, \ldots, m_p\}$, $m_1 < \cdots < m_p = m$, where $m$ is the full encoding dimension. We additionally maintain a core size $c$, $0 \leq c \leq m_1$, so that for each prefix $m_i \in \mathcal{M}$, we split $\mathbf{f}(\mathbf{x}) = \sigma\left(\mathbf{W}^{enc}\mathbf{x} + \mathbf{b}^{enc}\right) \in \mathbb{R}^m \longrightarrow \left[\mathbf{f}(\mathbf{x})_{0:c}, \mathbf{f}(\mathbf{x})_{c:m_i}\right]$, where

$$\mathbf{f}(\mathbf{x})_{0:c} \in \mathbb{R}^c \quad \text{(core)}, \quad \mathbf{f}(\mathbf{x})_{c:m_i} \in \mathbb{R}^{m_i-c} \quad \text{(non-core neurons)}. \tag{3}$$

We employ a tied decoder $\mathbf{W}^{dec} = (\mathbf{W}^{enc})^T$ to reconstruct the input:

$$\hat{\mathbf{x}}_i = \mathbf{W}^{dec}_{0:m_i} \left[\mathbf{f}(\mathbf{x})_{0:c}; \mathbf{f}(\mathbf{x})_{c:m_i}\right] + \mathbf{b}^{dec}, \tag{4}$$

where the extra block $\mathbf{z}_i^{extra} = \mathbf{f}(\mathbf{x})_{c:m_i}$ undergoes sparsification. Once neurons transition into the core, their encoding weights $\mathbf{W}^{enc}_{0:c}$ and biases $\mathbf{b}^{enc}_{0:c}$ become fixed, whereas all decoding weights, including those corresponding to the core, remain trainable.

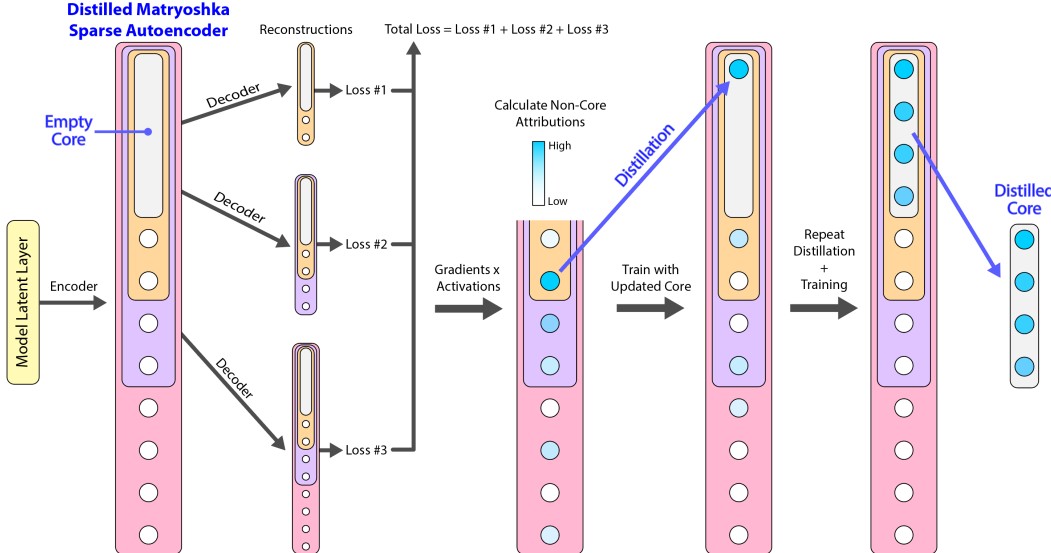

Figure 1: Schematic illustration of the Distilled Matryoshka Sparse Autoencoder (SAE). A model's latent layer is encoded into a nested sparse latent structure with an initially empty "core." This core is exempt from the sparsity constraint (BatchTopK) applied to outer neurons. Each nested sparse latent subset independently reconstructs the input through shared decoder layers, contributing separately to the total reconstruction loss. After inserting the Distilled Matryoshka SAE into the original model, we apply Integrated Gradients to compute each latent neuron's contribution to the model's outputs and select the highest attribution neuron within the most nested sparse layer. During distillation, this neuron's encoding weights are frozen and integrated into the core, while its decoding weights remain trainable. Iterative repetition of this distillation process progressively builds a compact and interpretable set of monosemantic features.

### 3.1.1 BATCHTOPK ACTIVATION ON NON-CORE NEURONS

To impose sparsity exclusively on the non-core neurons, we utilize the BatchTopK activation proposed in Bussmann et al. (2024). Given a mini-batch $\{x_j\}_{j=1}^B$, we form $\mathbf{Z}^{extra} = \left[ f(x_j)_{c:m_i} \right]_{j=1}^B \in \mathbb{R}^{B \times (m_i - c)}$,

flattening it into a vector of length $B \times (m_i - c)$. We then determine the threshold $\tau$, corresponding to the $(B \times K)$-th largest activation in the batch. Thus, sparsification is applied as:

$$\mathbf{z}_i^{extra} = \text{BatchTopK}\left(\mathbf{Z}^{extra}; K\right) = \mathbf{Z}^{extra} \odot \mathbf{1}\left[\mathbf{Z}^{extra} \geq \tau\right], \tag{5}$$

retaining an average of $K$ active neurons per sample in the non-core neurons, while leaving core activations dense. Choice of K, batch size B, and attribution interval T is detailed in Appendix A.2, along with the grid we swept and how we picked defaults.

### 3.1.2 TRAINING OBJECTIVE

Our training objective employs a mean-squared reconstruction loss, defined as:

$$\mathcal{L}(\mathbf{x}) = \sum_{m \in \mathcal{M}} \left\| \mathbf{x} - \left(\mathbf{f}(\mathbf{x})_{0:m} \mathbf{W}_{0:m}^{\text{dec}} + \mathbf{b}^{\text{dec}}\right) \right\|_2^2, \tag{6}$$

encouraging each nested latent representation (core plus non-core neurons) to faithfully reconstruct the input.

### 3.1.3 ATTRIBUTION-GUIDED CORE GROWTH

To progressively grow the core using attribution guidance, we periodically (after each training cycle) evaluate neuron importance on a sampled set of token positions. We restrict candidates to the first non-core Matryoshka block $S = [c : c+m_1]$. For each candidate latent $i \in S$ we compute a Grad×Activation score against the LM next-token cross-entropy, aggregated by a high quantile:

$$\alpha_i = \text{quantile}_q\left(\left|\text{ReLU}((\mathbf{w}_i^{enc})^\top \mathbf{x}_{b,t} + b_i^{enc}) \cdot \left(\nabla_{\mathbf{x}_{b,t}}\mathcal{L}_{\text{LM}} \cdot \frac{\mathbf{W}_{:,i}^{dec}}{\|\mathbf{W}_{:,i}^{dec}\|}\right)\right|\right)_{(b,t)\in S_{\text{tok}}}, \quad i \in [c : c+m_1),$$

(7)

where $\mathbf{x}_{b,t} \in \mathbb{R}^d$ is the LM residual at the chosen hook, $S_{\text{tok}}$ is the sampled set of token positions, and $\mathbf{W}_{:,i}^{dec}/\|\mathbf{W}_{:,i}^{dec}\|$ is the unit decoder direction for latent $i$. We then sort $\{\alpha_i\}_{i\in S}$ in descending order and promote the smallest prefix $P$ whose cumulative sum reaches a target fraction $\rho$ of the total attribution, i.e., $\sum_{i\in P} \alpha_i \geq \rho \sum_{j\in S} \alpha_j$. All latents in $P$ are appended to the core $\mathcal{C}$ (preserving the existing core order). Their encoder rows (weights and biases) are frozen in the next training cycle, while BatchTopK gating (with a per-example $K$-th order-statistic threshold) continues to apply only to the remaining non-core latents. Repeating this cycle yields a concise, monosemantic core nested within a Matryoshka arrangement trained under the unified reconstruction objective. For a step-by-step walkthrough of the full training and distillation loop, see Appendix A.1.

## 4 EXPERIMENTS

We evaluated the DMSAE on the SAEBench suite Karvonen et al. (2025), first focusing on how core promotion size interacts with $L_0$ sparsity. We began with a sweep over sparsity levels ($L_0 \in \{20, 40, 80, 160, 320, 640\}$) and promotion fractions (selecting the top $\rho \in \{10, 30, 50, 70, 90\}\%$ of attribution from the smallest Matryoshka prefix). We trained using each set of parameters for 100M tokens and reporting SAEBench metrics in Fig. 2. We then quantified how $\rho$ and the BatchTopK budget $k$ translate into the number of neurons promoted to the core (Fig. 3). Guided by this analysis, we chose a $L_0$=20 and $\rho$=90% to train longer to 500M tokens and compared this DMSAE against the Matryoshka baseline and other SAE variants (BatchTopK, TopK, JumpReLU) across SAEBench benchmarks in Fig. 4.

**SAEBench behavior across sparsity.** Figure 2 evaluates DMSAEs trained with promotion fractions of 10, 30, 50, 70, and 90 percent of the smallest Matryoshka prefix, using the corresponding pre-trained SAEBench models Karvonen et al. (2025) of various $L_0$ values. Models were trained using 100 million tokens. DMSAE models were evaluated using benchmarks from SAEBench Karvonen et al. (2025). We observed small but consistent improvements at lower values of $L_0$, while larger $L_0$ values led to benchmarks falling below the Matryoshka baseline. Changing the promotion fraction alters the curves only slightly, and increasing the fraction did not appear to introduce low quality neurons into the core. A plausible explanation for the drop at high $L_0$ is that looser sparsity allows the non-core neurons to overlap more with the core, which can make optimization harder and impair benchmark evaluation. To further investigate this, we analyzed how attribution coverage $\rho$ and BatchTopK $k$ translate into core size (Fig. 3). Empirically, we find that 90% of the total attribution is captured by promoting only 8.7% of the smallest-prefix neurons ($m_1$=2048), implying limited reconstruction importance among the remaining non-core neurons. Together, these trends suggest that prioritizing values that reach a target attribution sum using as few neurons as possible may be optimal. Selecting $\rho$=90% with $k$=20 therefore results in a minimal core size while preserving dictionary capacity for retraining with 500 million tokens (Fig. 4).

**DMSAE performance after 500M tokens.** With $k$=20 and a 90% promotion fraction, training the DMSAE with 500M tokens surpassed the Matryoshka baseline on four of six SAEBench metrics: AutoInterp, SCR, Sparse Probing, and Absorption while trailing modestly on Loss-Recovered and RAVEL. These gains are achieved while freezing only $\sim 0.3\%$ of the dictionary into the core, indicating that attribution-guided distillation can improve performance with minimal trade-off. Extended training and additional distillation rounds should accumulate high attribution, monosemantic features in the core.

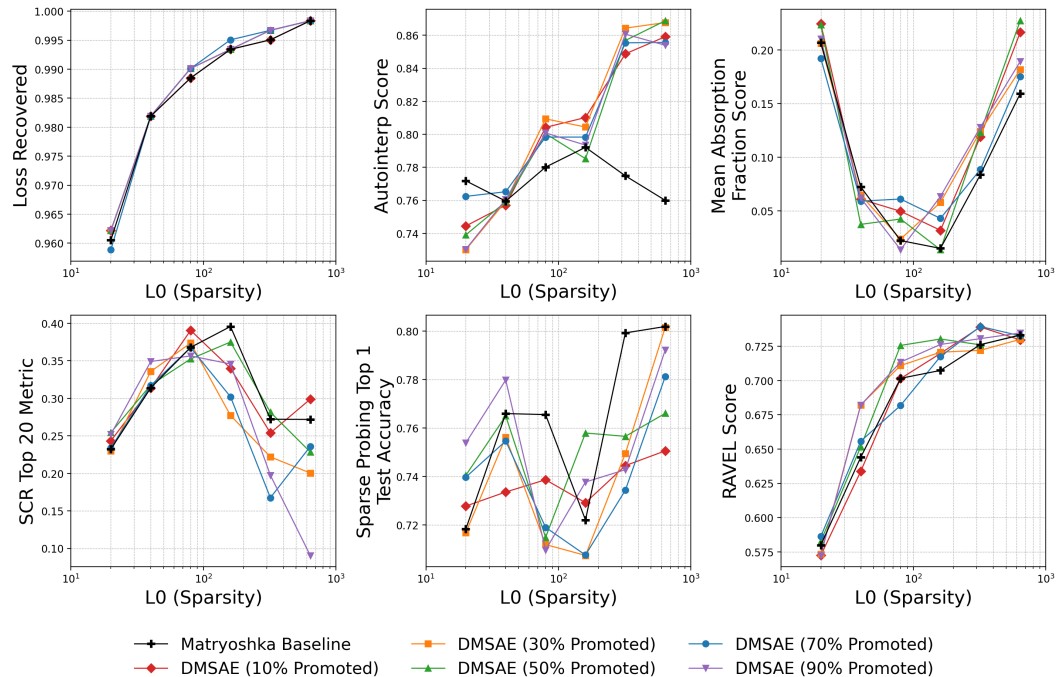

Figure 2: **SAEBench metrics vs. sparsity ($L_0$).** We compare Matryoshka and DMSAE variants across six SAEBench metrics while sweeping $L_0$. DMSAEs were trained with promotion fractions of 10, 30, 50, 70, and 90 percent from the smallest prefix of an Matryoshka SAE dictionary of size 65k. Small $L_0$ such as $k=20$ shows modest gains, whereas higher $L_0$ values at or above 80 perform worse than the baseline. The promotion fraction has only a minor effect, so we adopt 90 percent for subsequent experiments in Fig. 4.

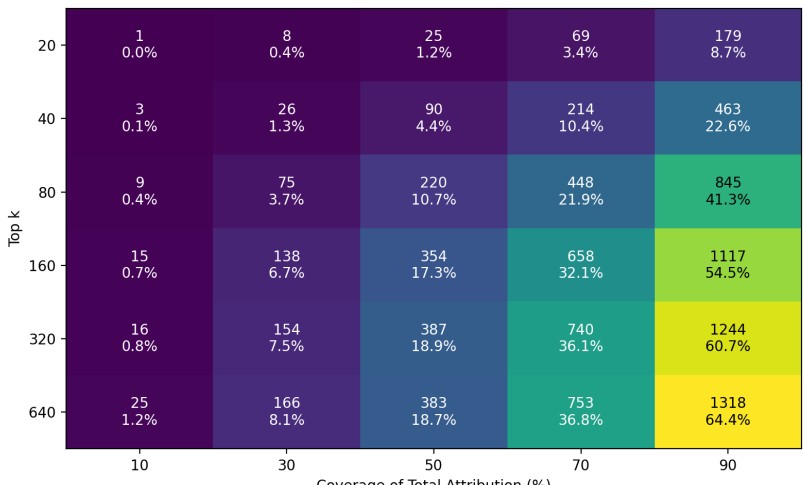

Figure 3: **Core size as a function of attribution coverage and sparsity.** Heatmap of the number of neurons promoted to the core under a grid of attribution coverage targets (columns; fraction of total attribution $\rho \in \{10, 30, 50, 70, 90\}\%$) and BatchTopK sparsity budgets (rows; top-$k \in \{20, 40, 80, 160, 320, 640\}$). Numbers indicate the absolute count of promoted neurons, and the percentages (on the next line) indicate the fraction of the smallest prefix ($m_1=2048$) that is promoted. Higher coverage thresholds and larger $k$ produce larger cores, with diminishing returns at very large $k$. We use this sweep to select $\rho=90\%$ at $k=20$ for subsequent experiments.

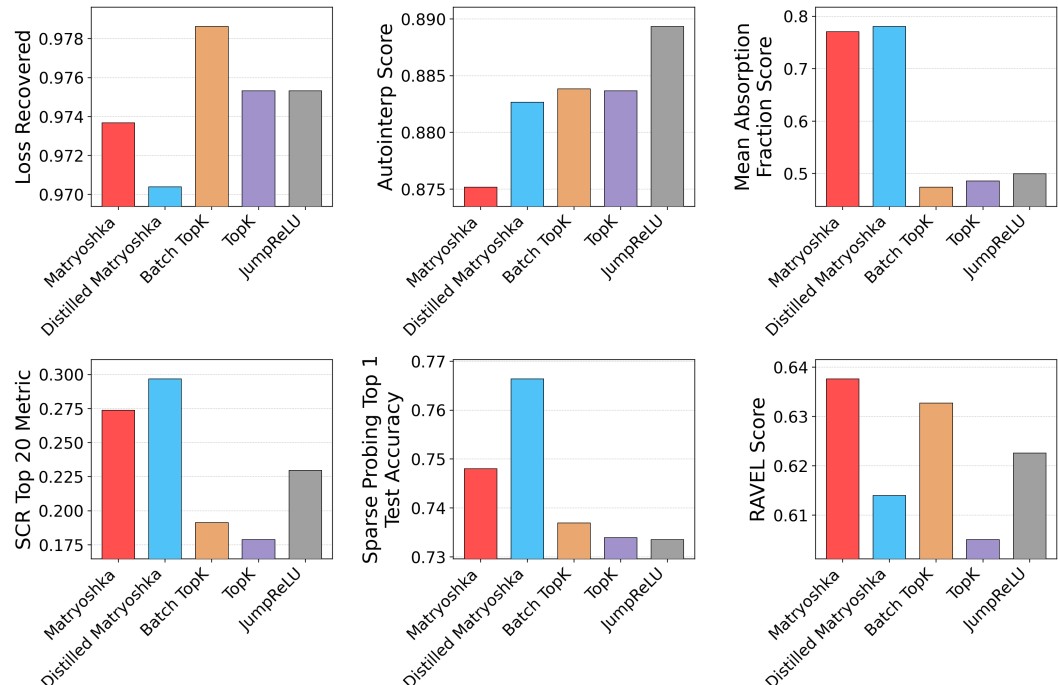

Figure 4: **Fixed operating point comparison at** $L_0=20$. DMSAE with 90 percent promotion from the smallest Matryoshka prefix, trained to 500 million tokens, compared with Matryoshka, Batch TopK, TopK, and JumpReLU. DMSAE improves over the Matryoshka baseline on AutoInterp, Absorption where lower is better, SCR Top 20, and Sparse Probing Top 1, but is slightly behind on Loss Recovered and RAVEL.

## 5 LIMITATIONS

The main limitation of the DMSAE method is computational: each training cycle (train→attribution→promote→retrain) costs the same as training the original baseline Matryoshka SAE. Additional limitations include reliance on imperfect attribution signals and the potential to lock-in poor quality neurons from early "mispromotions". However, these constraints are orthogonal to the main DMSAE concept and can be addressed with different engineering choices, depending on the model. Finally, our evaluation focuses on analyzing one layer and model with SAEBench metrics, other causal test and user studies remain to be done once additional training cycles are completed in the future.

## 6 DISCUSSION AND CONCLUSION

We have introduced Distilled Matryoshka Sparse Autoencoder, an attribution-guided framework that incrementally builds a frozen core of monosemantic neurons within a Matryoshka SAE. By freezing encoder weights of the most salient features and allowing subsequent training to optimize decoder weights, our method can distill a dense core that is monosemantic and interpretable. Future work will examine smaller, domain-specific models in physics and biology to test whether repeated DMSAE cycles can yield distilled cores that can independently fully reconstruct a model layer. Overall, Distilled Matryoshka SAEs offer a scalable, attribution-guided path to dense and hierarchically structured dictionaries that may help mitigate limitations of sparse autoencoders.

---

**Algorithm 1** DMSAE: Training Loop & Attribution-guided Distillation

---

**Require:** Dataset $\mathcal{D}$,
 LM activations $\{\mathbf{x}_j\}_{j=1}^N$,
 core set $\mathcal{C}$ of size $c$, sparsity $K$,
 nested sizes $\mathcal{M} = \{m_0, \dots, m_p\}$ with $m_p = m$, $m_0 = c$,
 Matryoshka SAE training epochs $T$, batch size $B$, learning rate $\eta$,
 attribution quantile $q \in (0, 1)$, promotion fraction $\rho \in (0, 1]$
**Ensure:** Encoder $\{\mathbf{W}^{enc}, \mathbf{b}^{enc}\}$ (with $c$ rows frozen)
 **Phase I: Train Distilled Matryoshka SAE**
 **for** epoch = 1 to $T$ **do**
  **for** each mini-batch $\{\mathbf{x}_j\}_{j=1}^B \sim \mathcal{D}$ **do**
   $\mathbf{z} = \mathbf{f}(\mathbf{x}_{1:B}) = \mathrm{ReLU}(\mathbf{W}^{enc}\mathbf{x}_{1:B} + \mathbf{b}^{enc}) \in \mathbb{R}^{B \times m}$
   $\mathbf{z}_{:, c:m}^{\mathrm{extra}} \leftarrow \mathrm{BatchTopK}(\mathbf{z}_{:, c:m}; K) = \mathbf{z}_{:, c:m} \odot \mathbf{1}[\mathbf{z}_{:, c:m} \geq \tau]$
   $\mathcal{L}_{\mathrm{SAE}}(\mathbf{x}_{1:B}) = \sum_{m' \in \mathcal{M}} \left\| \mathbf{x}_{1:B} - (\mathbf{z}_{:, 0:m'}\mathbf{W}_{0:m'}^{\mathrm{dec}} + \mathbf{b}^{\mathrm{dec}}) \right\|_2^2$
   backpropagate and update unfrozen weights
  **end for**
 **end for**
 **Phase II: Attribution-guided Distillation from first non-core block**
 Let the candidate index range be $S = [c : c+m_1)$ (the first non-core block).
 Sample a set of token positions $S_{\mathrm{tok}} \subseteq \{(b, t)\}$, compute next-token CE loss $\mathcal{L}_{\mathrm{LM}}$ and its gradient
 $g_{b,t} = \nabla_{x_{b,t}} \mathcal{L}_{\mathrm{LM}}$.
 **for** each $i \in S$ **do**
  $u_i \leftarrow \mathbf{W}_{:,i}^{dec} / \|\mathbf{W}_{:,i}^{dec}\|$
  For each $(b, t) \in S_{\mathrm{tok}}$: $z_i(b, t) = \mathrm{ReLU}(w_i^{enc} \cdot x_{b,t} + b_i^{enc})$, $s_i(b, t) = |z_i(b, t) \cdot (g_{b,t} \cdot u_i)|$
  $\alpha_i \leftarrow \mathrm{quantile}_q(\{s_i(b, t) : (b, t) \in S_{\mathrm{tok}}\})$
 **end for**
 Sort $i \in S$ by descending $\alpha_i$; let $A_{\mathrm{tot}} = \sum_{i \in S} \alpha_i$.
 Choose the smallest prefix $P$ such that $\sum_{i \in P} \alpha_i \geq \rho A_{\mathrm{tot}}$.
 $\mathcal{C} \leftarrow \mathcal{C} \cup P$
 **for** each $d \in \mathcal{C}$ **do**
  freeze encoder weights $\mathbf{W}_d^{enc}$ and bias $\mathbf{b}_d^{enc}$ (applied in the next training cycle)
 **end for**

---

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

## A  SOFTWARE AND DATA

Code Availability: Code and data will be made available upon publication

## B  USE OF LLMS

We used LLMs to assist with LaTeX formatting, light copy editing, and code generation.

