# OpenReview forum: "Attribution Guided Distillation for Matryoshka Sparse Autoencoders"
_ICLR.cc/2026/Conference — ICLR 2026 Conference Withdrawn Submission_

### Official Review · Reviewer_Ndtg · 2025-10-23

**Soundness:** 1
**Presentation:** 1
**Contribution:** 2
**Rating:** 2
**Confidence:** 4

**Summary:**

This paper proposes Distilled Matryoshka SAEs (DMSAEs), an extension of Matryoshka SAEs (MSAEs). The method first trains an MSAE, then uses gradient-based attribution to identify important latents in the smallest prefix. These latents are promoted into a frozen "core," which is exempt from the sparsity penalty (BatchTopK) applied to other features. The authors evaluate this approach on layer 12 of Gemma-2-2B, comparing it against MSAE baselines on the SAEBench suite, and claim improvements

**Strengths:**

See below

**Weaknesses:**

See below

**Questions:**

**Overall Rating:**
I am not comfortable recommending acceptance for this paper. The core idea of an attribution-guided distillation process is interesting, but the work in its current form is undermined by key missing details, and what I believe to be a critical methodological flaw in its experimental comparison (not including the latents in the core in the L0 calculation). The evidence provided is insufficient to conclude that the proposed method offers a genuine improvement over existing techniques. I'm open to increasing my score substantially if these issues are addressed.

### Major Comments

The following are things that, if adequately addressed, would increase my score.

1. **The L0 comparison is critically flawed and likely invalidates the main results.** My primary concern is that the evaluation does not account for the additional active features in the DMSAE's core. The core is exempt from the BatchTopK penalty (Lines 191-193, 206-215), yet the L0 plots (Fig. 2, Fig. 4) report identical, discretized sparsity levels for all methods. This strongly suggests that L0 is only being measured on the non-core latents. If so, the DMSAE's effective L0 is actually K + c (non-core sparsity + core sparsity), while the baseline's is just K. This is a major confounder, as it is well-established that SAE performance improves with higher L0. The fact that core latents are frozen improves this somewhat, but it's still an unfair comparison. The claimed gains may simply be an artifact of DMSAE using more features. For a fair comparison, you must re-run your experiments matching the total effective L0 across all methods.
2. **Missing Crucial Experimental Details.**
The paper omits several key details required for reproducibility and a full assessment of the methodology.

	- **Distillation Loop:** How many train -> promote -> retrain cycles were performed for the final models in Figure 4?
	- **Re-initialization:** When features are promoted to the core, are the weights of the remaining non-core latents re-initialized or do they continue training from their previous state?
	- **Architecture:** What is the total dictionary width (m)?
	- **Baselines:** Where do the baseline models in Figure 4 come from? Were they trained from scratch for this paper, or are they pre-trained models from SAEBench?
3. **Lack of a Compute-Matched Comparison.**
The paper acknowledges in the limitations that each DMSAE cycle "costs the same as training the original baseline Matryoshka SAE" (lines 354-356). If the final DMSAE model required multiple cycles, its total training compute could be much more than the baseline. Training FLOPs improve performance, so this is another major confounder. A fair comparison requires training the MSAE baseline for an equivalent number of FLOPs to see if its performance improves and closes the gap. Without this, it's impossible to distinguish algorithmic improvements from the brute-force benefits of more training.
4. **Interpretability Claims Are Unsubstantiated.**
The paper's primary motivation is to distill a "monosemantic and interpretable" core (line 360), but it offers no direct evidence that the promoted features are, in fact, interpretable. High-attribution features are not necessarily monosemantic; they could be high-frequency, polysemantic features that are crucial for reconstruction but semantically messy. While they are frozen after promotion to the core, methods like TopK can often find seemingly uninterpretable, high frequency latents. Suggested experiments:
	- An analysis of the **activation frequency** of the core latents.
	- **Frequency-weighted auto-interpretability scores** to ensure high scores aren't driven by a few clean but rare features.
	- A small-scale **human interpretability study** on the core features to sanity-check their quality.

### Minor Comments

The following are suggestions that I hope will improve the paper, but are unlikely to change my score.

- The attribution-guided promotion is the core technical contribution, but it lacks a simple ablation. It would be valuable to compare against a baseline that promotes features based on a simpler heuristic, such as **activation frequency**, to demonstrate that the gradient-based attribution is providing value.

I found this work somewhat difficult to follow, so apologies if I have missed or misunderstood any key details. Please correct me if I have!

---

> ### Author Response · Authors · 2025-12-01
> **Withdrawal**
>
> We thank the reviewer for their thoughtful comments and suggestions. Their feedback has highlighted several important improvements that go beyond the scope of a revision, so we have decided to withdraw the paper. Their input will be very helpful as we develop our work further.

---

### Official Review · Reviewer_KLH6 · 2025-10-30

**Soundness:** 3
**Presentation:** 2
**Contribution:** 2
**Rating:** 6
**Confidence:** 4

**Summary:**

This paper proposes Distilled Matryoshka Sparse Autoencoders (DMSAEs), an attribution-guided extension of Matryoshka SAEs for extracting monosemantic features from LLM activations. The method iteratively identifies neurons with high attribution to the model’s next-token loss and freezes them into a permanent “core” shared across all Matryoshka prefixes. The goal is to distill a compact and interpretable latent basis that avoids redundancy and feature fragmentation. The authors evaluate on Gemma-2-2B layer 12 using SAEBench and report modest but consistent improvements in interpretability metrics over standard MSAEs. The paper is conceptually interesting and technically competent, though empirically limited and computationally heavy.

**Strengths:**

The attribution-guided distillation idea is elegant and grounded in a clear causal motivation, namely selecting features that matter for model behavior, not just for reconstruction. The design integrates attribution analysis with hierarchical feature learning in a way that directly addresses known SAE pathologies such as feature absorption and redundancy. The experimental methodology is transparent and uses established SAEBench benchmarks, showing reproducible gains in interpretability at minimal cost in reconstruction fidelity.

**Weaknesses:**

The empirical scope is narrow, limited to a single layer of one model. The attribution signals used for core promotion are noisy and their semantic validity is unclear, high gradient attribution does not necessarily imply human-interpretable meaning or actual causal relevance, but it is probably an okay approximation. The method is computationally expensive, requiring multiple full training cycles, which undermines its scalability to larger models or multilayer analysis. The evaluation only shows limited improvement over baselines and is limited to a single layer, which make this hard to judge as SAE Bench results can differ significantly by layer. Overall the paper’s promise exceeds the evidence currently presented.

**Questions:**

- How stable are the promoted “core” neurons across different random seeds, datasets, or layers?
- Can the authors provide qualitative examples of distilled features that illustrate improved monosemanticity beyond benchmark scores?
- What is the computational cost per distillation cycle in practice, and are there strategies to improve that for larger-scale applications?

---

> ### Author Response · Authors · 2025-12-01
> **Withdrawal**
>
> We thank the reviewer for their thoughtful comments and suggestions. Their feedback has highlighted several important improvements that go beyond the scope of a revision, so we have decided to withdraw the paper. Their input will be very helpful as we develop our work further.

---

### Official Review · Reviewer_af9R · 2025-11-01

**Soundness:** 2
**Presentation:** 2
**Contribution:** 2
**Rating:** 4
**Confidence:** 4

**Summary:**

The paper proposes Distilled Matryoshka SAEs. Starting from a standard Matryoshka sparse autoencoder, the method periodically computes an attribution score, then "promotes" the highest attribution neurons into a frozen core that is shared by all prefixes. Encoder rows for promoted neurons are frozen, decoder weights remain trainable, and BatchTopK sparsity is applied only to the remaining non core latents. Attribution uses a GradInput projected onto the unit decoder direction and aggregated across token positions with a high quantile; promotion selects the smallest prefix of neurons that reaches a target fraction of total attribution. Proof of concept experiments at layer twelve of Gemma two two B report small but consistent gains over a Matryoshka baseline on four of six SAEBench metrics at low $\ell_0$, with weaker results at higher $\ell_0$.

**Strengths:**

The paper reads well, here are in my opinion the positive points (P) of the article:

P1. The idea of a small, dense, reusable core shared by all prefixes is clear and easy to reason about, and Algorithm 1 makes it implementable within existing Matryoshka code paths. The schematic on page four is helpful although a bit complex.

P2. The attribution definition is explicit. Equation 7 specifies Grad times Activation projected onto the decoder direction, with a quantile over token positions, I believe i could reproduce the ranking used for promotion.

P3. The empirical sweep is at least systematic at low sparsity, and the heatmap on page six relating core size to attribution coverage and sparsity is interesting!

**Weaknesses:**

In my opinion, this paper is interesting but has some weaknesses thta I will describe here, I will group them in major (M) and minor (m) problems.

M1. Method complexity versus modest gains. The pipeline adds attribution computation, ranking, promotion, and retraining. Each cycle costs roughly one full Matryoshka training according to Section 5, yet Figures 2 and 4 show improvements that are small and not universal, with drops at higher $\ell_0$ and slight deficits on Loss Recovered and RAVEL even at the best operating point. Without stronger gains or a clearer story for when the method helps, the cost benefit picture is not compelling.

M2. Missing baselines. MP-SAE [1] and collaborators, as well as orthogonal matching pursuit style encoders, create a conditional selection effect that is highly relevant to absorption and composition. They should appear in the benchmark and in related work. The absence makes it hard to tell whether your benefit is coming from conditional selection like behavior or from the distillation loop.

[1] From Flat to Hierarchical: Extracting Sparse Representations with Matching Pursuit, Costa & al.

M3. Neuron basis attribution versus directional derivatives at the dictionary level. The paper ranks individual latent neurons using Grad times Activation projected on the unit decoder. This is close to a directional derivative along decoder directions, but the exposition leaves open alternatives that might better match the objective. For example, rank by the directional derivative of the language model loss with respect to the reconstruction vector after projecting the gradient onto the full dictionary span, or decompose the gradient into the current active set and measure marginal attributions conditional on co active atoms? Can you expand on that and if possible do a short comparison?

M4. Why use a high quantile aggregator. Equation 7 aggregates per token scores with a quantile but gives limited motivation beyond robustness. Can you justify the choice of q? compare to an average, a trimmed mean? And if possilble show stability of the ranking across batches. Sensitivity of the promoted set to q should be reported, since Figure 2 suggests limited effect from promotion fraction while the aggregator might still change which neurons enter the core.

M5. Related work is thin for ICLR expectations. The background is largely limited to recent SAE variants and Matryoshka works and contain 17 citations. This undercuts positioning and novelty arguments.

M6. Ambiguity in attribution definition across the paper. Figure 1 caption refers to Integrated Gradients, while the method and Algorithm 1 actually implement a GradientInput quantile. This must be made consistent and the reason for the final choice explained.

Now for the minor problems:

m1. Can you clarify notation and normalisation in Equation 7? Also state whether decoder columns are normalised continuously during training and whether the ReLU term uses pre or post normalisation activations?

m2. Fonts are really small in the six panels of Figure 2 and in the heat map of Figure 3. Also if possible can you add uncertainty bands or error bars to all panels and ensure captions state the exact operating point and the absolute deltas.

m3. Can you explain the choice to restrict candidates to the first non core block? A short ablation that allows selection across deeper prefixes would test whether important features are being excluded by construction.

m4. Writing and structure. Several sections read densely and the related work omits key areas as noted. A careful pass to streamline Section 3 and to broaden Section 2 would help.

**Questions:**

See the Major points M1-6 and the minor points 1-4.

---

> ### Author Response · Authors · 2025-12-01
> **Withdrawal**
>
> We thank the reviewer for their thoughtful comments and suggestions. Their feedback has highlighted several important improvements that go beyond the scope of a revision, so we have decided to withdraw the paper. Their input will be very helpful as we develop our work further.

---

### Note · Authors · 2025-12-01

I have read and agree with the venue's withdrawal policy on behalf of myself and my co-authors.